# Risk factors for psychiatric symptoms in patients with long COVID: A systematic review

**Halwa Zakia, Kent Pradana, Shelly Iskandar** *

Faculty of Medicine, Department of Psychiatry, Padjadjaran University, Hasan Sadikin General Hospital, Bandung, Indonesia

* shelly_bdg@yahoo.com

**Data Availability Statement:** All relevant data are within the paper and its Supporting Information files.

## Abstract

Prolonged symptoms of COVID-19 have been found in many patients, often known as Long COVID. Psychiatric symptoms are commonly seen in Long COVID patients and could last for weeks, even months, after recovery. However, the symptoms and risk factors associated with it remain unclear. In the current systematic review, we provide an overview of psychiatric symptoms in Long COVID patients and risk factors associated with the development of those symptoms. Articles were systematically searched on SCOPUS, PubMed, and EMBASE up to October 2021. Studies involving adults and geriatric participants with a confirmed previous COVID-19 diagnosis and reported psychiatric symptoms that persist for more than four weeks after the initial infection were included. The risk of bias was assessed using the Newcastle-Ottawa Scale (NOS) for observational studies. Prevalence rates and risk factors associated with psychiatric symptoms were collected. This present study was registered at PROSPERO (CRD42021240776). In total, 23 studies were included. Several limitations in this review were the heterogeneity of studies' outcomes and designs, studies limited to articles published in English, and the psychiatric symptoms mainly were assessed using self-report questionnaires. The most prevalent reported psychiatric symptoms, from the most to the least reported, were anxiety, depression, post-traumatic stress disorder (PTSD), poor sleep qualities, somatic symptoms, and cognitive deficits. Being female and having previous psychiatric diagnoses were risk factors for the development of the reported symptoms.

## Introduction

On 8[th] December 2019, an acute respiratory disease named Corona Virus Disease 2019 (COVID-19) was found in Wuhan City, Hubei Province, China [1]. Globally, as of 1[st] November 2021, there had been 245 million COVID-19 cases worldwide [2]. COVID-19 is highly transmittable and caused by severe acute respiratory syndrome coronavirus 2 (SARS-CoV-2) [3].

Although the main reported symptom is acute respiratory distress syndrome, COVID-19 also affects other organs, including the brain [4]; and affects mental health. COVID-19 contributes to psychiatric symptoms and disorders, mainly due to extreme fear, anxiety, and

**Funding:** The author(s) received no specific funding for this work.

**Competing interests:** The authors have declared that no competing interests exist.

negative social behaviours, leading to distress reactions, health risk behaviours and mental health disorders [5]. A systematic review in 2021 showed that depression, anxiety, post-traumatic stress disorder (PTSD), cognitive deficits, fatigue, and sleep disturbances were commonly found in COVID-19 survivors [6]. Studies also found other psychiatric symptoms such as sleep disturbance, somatization, obsessive-compulsive, phobic anxiety, and hostility among COVID-19 participants that persisted for a long time after discharge. These persistent psychiatric symptoms may require urgent attention in terms of COVID-19 treatment [7].

The symptoms of prolonged COVID-19 sequel and its complications are generally known as ongoing symptomatic COVID-19, post COVID-19 syndrome, and "Long COVID" [8]. A guideline by National Institute for Health and Care Excellence (NICE) differentiates between ongoing symptomatic COVID-19, defined as having symptoms for 4 to 12 weeks, and post COVID-19 syndrome, defined as having symptoms for more than 12 weeks. Long COVID-19 includes both ongoing symptomatic COVID-19 and post COVID-19 syndrome, which means Long COVID-19 is defined by an individual with persistent symptoms for more than four weeks after the initial COVID-19 diagnosis [8].

Preliminary data suggested that some psychiatric disorders, such as anxiety and depression, persisted in patients who had recovered from COVID-19 [9]. Around 30% of COVID-19 patients with a negative virologic test are still experiencing depression and anxiety [10]. The most common psychiatric disorders among COVID-19 patients include depression, anxiety, thoughts of self-harm, and PTSD [11]. These psychiatric symptoms become a burden for patients with Long COVID-19, especially because COVID-19 is still considered a global pandemic and still infects many people worldwide.

Prolonged psychiatric illness due to COVID-19 is associated with persistent physical symptoms, such as breathlessness and myalgia [12]. Consequently, studies about prolonged psychiatric symptoms of COVID-19 are needed to give further explanations about Long COVID-19 related symptoms [12]. Although a systematic review regarding psychiatric complications among post COVID-19 patients had been done before [6], a review focusing on psychiatric symptoms and factors associated with it among Long COVID-19 patients is still needed to develop new strategies in terms of mental illness prevention in Long COVID-19 patients. Therefore, this review focused on psychiatric symptoms, specifically among patients with Long COVID, to provide an overview of psychiatric symptoms in Long COVID-19 patients and risk factors associated with the development of those symptoms by using the most recent literature.

## Materials and methods

### Eligibility criteria

Studies were selected if they met the following criteria: published in English and full-text was available; published between January 2020 and October 2021. The inclusion criteria included the following: the study design was either a cohort study, case-control study, cross-sectional study, or case series; participants were adults and geriatrics with a confirmed previous COVID-19 infection; and studies that reported psychiatric symptoms that persisted after four weeks since the initial infection and its associated factors. The exclusion criteria were review articles and studies that included children or adolescents.

### Guidelines

The present study was registered at PROSPERO (CRD42021240776). This systematic review was reported by following the guidelines by Preferred Reporting Items for Systematic Reviews and Meta-Analysis (PRISMA) [13].

## Search strategy

Literature search was performed on Scopus, Embase, and PubMed on 24[th] October 2021. Pre-defined search items included multiple combinations of the following: ("psych*" OR "mental" OR "depression" OR "anxiety" OR "PTSD") AND ("Long-COVID" OR "Long Haulers" OR "Post-acute COVID" OR "Persistent COVID" OR "Post COVID" OR "Long-term COVID"). Studies obtained from the search were transferred into the Excel database, and duplicates were removed. We searched reference lists and carried out citation searching for included papers and previous reviews in this area.

## Data extraction

One reviewer searched the literature and extracted the data to an Excel database. The titles and abstracts were screened to determine eligibility by two reviewers. After excluding studies that were not eligible, a full-text review was done to obtain detailed information. Data extraction included: author, year, country, study design, population, sample size, age, ratio female to male, duration from initial COVID-19 diagnosis, the prevalence of psychiatric symptoms, assessment tools, significant risk factors with its effect size (odds ratio, relative risk, hazard ratio, or prevalence ratio) and non-significant risk factors.

## Quality assessment

Two authors independently assessed the risk of bias using the Newcastle Ottawa Scale (NOS) [14]. Studies were scored 4 for the selection of participants, 2 for comparability, and 3 for pre-dictor ascertainment and analysis [15]. Based on the NOS criteria, studies with lower than 5 stars were considered low quality; 5 to 7 stars, moderate quality; and more than 7, high quality. Any disagreements were resolved by discussion until a consensus was reached.

# Results

## Study selection

A total of 2,218 articles were obtained from 3 databases, as shown in Fig 1 and S1 Table. After 1,022 duplicates were removed, 1,196 articles remained. Then 1,131 articles were excluded after screening titles and abstracts, and 65 abstracts met the inclusion criteria. A further 47 articles were removed for the following reasons: the study did not discuss the risk factors associated with Long COVID symptoms (n = 23), no full text was available (n = 13), the study did not include any psychiatric diagnosis (n = 5), the study did not define the onset of the COVID-19 symptoms or did not include the initial time of COVID-19 diagnosis (n = 4), participants were children/adolescents (n = 1), and the population did not include post-COVID participants (n = 1). Five articles were included via citation searching. Therefore, 23 articles were included.

## Characteristics of included studies

All included studies were published in 2021 except for 1 study, which was published in 2020. Fifteen studies were cohort, five were cross-sectional, and 3 were case-control. All 23 studies evaluated risk factors associated with psychiatric symptoms in Long COVID participants. The studies were conducted in thirteen different countries: five studies were from Spain, four from Italy, three from the UK, two from China, two from India, and one study from each of the following countries (Peru, Turkey, Poland, Switzerland, US, France, and Pakistan). The number of participants in each study varied from 48 to 273,618. The general characteristics of reviewed articles are listed in Table 1.

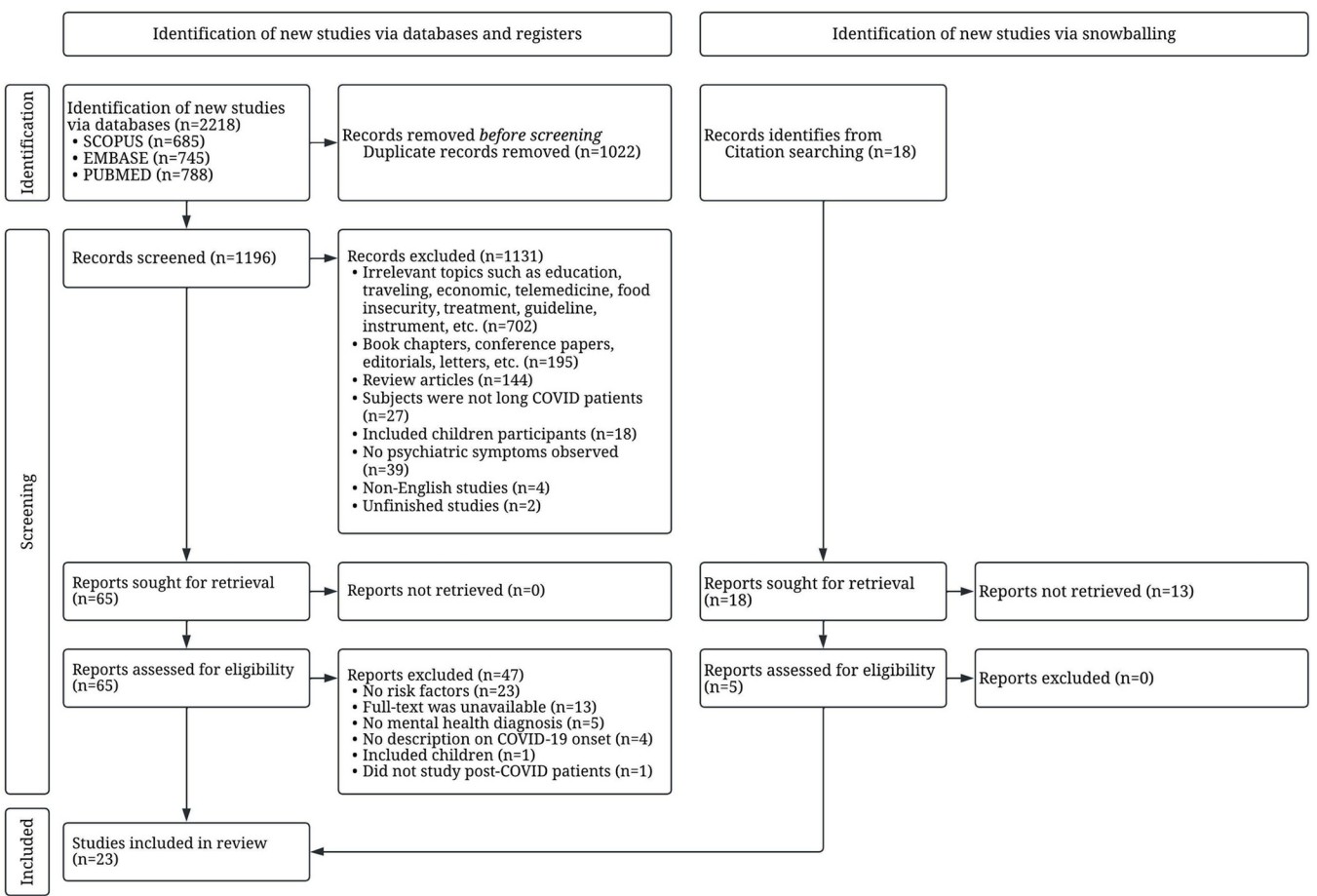

**Fig 1. Study selection using PRISMA flow diagram.**

## Methodological quality of the included studies

The overall qualities of reviewed articles are listed in Tables 1 and S2. The range of NOS in this review is 5 to 8. Three studies scored 5, nine studies scored 6, nine studies scored 7, and two studies scored 8. The average assessment score was 6.43 (moderate quality). The main problem in the article quality included in this review was the lack of sample size justification. Whereas representativeness of the sample, ascertainment of the exposure, and assessment of the outcomes were all clearly defined in the majority of included articles. The aim was clearly stated for almost all of the studies. The cross-sectional design of the included studies is vulnerable to three types of bias: selection, information, and confounding bias. Most research use convenience sampling, increasing the possibility of bias in the selection process. Numerous studies also failed to provide information on participant response rates.

## Psychiatric symptoms in Long COVID participants

Several psychiatric symptoms were reported in the included studies. Anxiety was found in 19 of 23 articles, with prevalence ranging from 6.8% [24] to 47.8% [19]. Depression was found in 17 articles, with prevalence ranging from 4.4% [24] to 35.9% [32]. PTSD was found in 7 articles, with prevalence ranging from 13.0% [9] to 42.8% [27]. Poor sleep quality, sleep disturbances, and insomnia were found in 13 articles, with prevalence ranging from 4.4% [24] to

**Table 1. Characteristics of reviewed articles.**

| Authors | Study population (country) | Study design | Sex (F/M) | Age, mean or median | Follow up time | Prevalence of psychiatric symptom | Assessment tools | NOS |
|---|---|---|---|---|---|---|---|---|
| Huarcaya-Victoria et al., 2021 [16] | 318 (Peru) | Cross sectional | 122/196 | Median 53.1 (IQR 51.8–54.4) years | Mean 102.1 days | Depressive symptoms 30.9% Somatic symptoms 35.2% Anxious symptoms 31.1% PTSD symptoms 29.5% | PHQ-9; GAD-7; PHQ-15; IES-R | 7 |
| Fernández-de-Las-Peñas, Torres-Macho, et al., 2021 [17] | 88 (Spain) | Case control | 35/53 | Mean 52.0 (SD 14.5) years | 7.2 months after discharge | Depressive symptoms 13.6% Anxiety symptoms 15.9% Poor sleep quality 45.5% | HADS-A/D; PSQI | 7 |
| Tanriverdi et al., 2021 [18] | 48 (Turkey) | Cross sectional | 26/22 | Mean 39.2 (SD 7.9) years | 12 weeks from diagnosis | Anxiety 33.3% Depression 29.2% Poor sleep quality 50.0% | HADS-A/D; PSQI | 5 |
| Sykes et al., 2021 [19] | 134 (UK) | Cohort | 46/88 | Mean 59.6 (SD 14.0) years | Median 113 days (range 46–167) after discharge | Anxiety 47.8% Sleep disturbance 35.1% | EQ-5D-5L | 8 |
| Dankowski et al., 2021 [20] | 102 (Poland) | Cross sectional | 57/45 | Mean 56 (SD 18) years | Mean 56 days from diagnosis | Sleep disturbance 37.3% Anxiety 33.3% Depressive symptoms 16.7% | BDI; STAI | 5 |
| Taquet et al., 2021 [21] | 273,618 (UK) | Cohort | 152157/121461 | mean 46.3 (SD 19.8) years | within 6 months and in the 3 to 6 months after diagnosis | From day 1 to 6 months post discharge: anxiety/depression 22.82% From 3 months to 6 months post discharge: anxiety/depression 15.49% | ICD-10 | 6 |
| Menges et al., 2021 [22] | 431 (Switzerland) | Cohort | 214/217 | Median 47 (IQR 33–58) Years | 6–8 months after diagnosis | Depression symptoms 26% Anxiety symptoms 32% Stress symptoms 16% | DASS-21 | 7 |
| Alemanno et al., 2021 [23] | 87 (Italy) | Cohort | 25/62 | Mean 67.23 (SD 12.89) Years | 1 month after discharge | Cognitive deficits 73.2% Depression 33,9% PTSD 42.8% | MoCA; HDRS; DTS | 5 |
| Romero-Duarte et al., 2021 [24] | 797 (Spain) | Cohort | 369/428 | Mean 63.0 (SD 14.4) Years | 6 months after discharge | Depressive symptoms 4.4% Anxiety symptoms 6.8% Sleep disturbances 4.4% | Standardized questionnaire | 5 |
| Soni et al., 2021 [25] | 30 cases, 30 controls (India) | Case control | Case 9/21 Control 10/20 | Cases mean 53.9 years Control mean 46.2 years | NR | NR | PHQ-9, lab test | 5 |
| C. Huang et al., 2020 [26] | 1,733 (China) | Cohort | 836/897 | Median 57 (IQR 47–65) years | Median 186 (IQR 175–199) days | Sleep difficulties 26% Anxiety/depression 23% | Standardized Questionnaire | 7 |

(*Continued*)

**Table 1.** (Continued)

| Authors | Study population (country) | Study design | Sex (F/M) | Age, mean or median | Follow up time | Prevalence of psychiatric symptom | Assessment tools | NOS |
|---|---|---|---|---|---|---|---|---|
| Mazza et al., 2021 [9] | 226 (Italy) | Cohort | 149/77 | Mean 58.52 (SD 12.79) years | 3 months after discharge | Depression ZSDS 28% Depression BDI 9% PTSD IES-R 22% PTSD PCL 13% Anxiety 30% Obsessive compulsive 26% Insomnia 24% | IES-R; PCL-5; ZSDS; BDI-13; STAI-Y; WHIIRS; OCI | 8 |
| Bellan et al., 2021 [27] | 238 (Italy) | Cohort | 96/142 | Median 61 (IQR 50–71) years | 3–4 months after discharge | PTS 42.85% | IES-R | 6 |
| Fernández-de-Las-Peñas, Pellicer-Valero, et al., 2021 [28] | 1,969 (Spain) | Cohort | 915/1054 | Mean 61 (SD 16) years | 6–10 months after discharge | Anxiety 15.7% Depression 18.9% | HADS | 6 |
| De Lorenzo et al., 2021 [29] | 251 (Italy) | Cohort | 72/129 | Mean 61.8 years | 1 and 3 months after discharge | Anxiety 25.5% insomnia 25.5% PTSD 22.4% | IES-R, STAI-Y; WHIIRS | 7 |
| Shang et al., 2021 [30] | 796 (China) | Cohort | 392/404 | Median 62.0 (IQR 51.0–69.0) years | 6 months after discharge | Sleep disorder 23.2% | Standardized questionnaire | 7 |
| Frontera et al., 2021 [31] | 196 (US) | Cohort | 68/128 | Median 68 (IQR 55–77) | 6 months from onset of neurological symptoms | Anxiety 47% Depression 29% Sleep disturbances 35% | Neuro-QoL | 7 |
| Fernández-de-las-Peñas, Gómez-Mayordomo, Cuadrado, et al., 2021 [28] | 205 (Spain) | Cross sectional | 123/82 | Mean 55.5 (SD 14.0) | Mean 7.3 (SD 0.6) months after hospital discharge | Depression 23.0% Anxiety 14.2% Poor sleep quality 38.5% | HADS-A/D; PSQI | 5 |
| Zahoor et al., 2021 [32] | 65 (Pakistan) | Cohort | 63/1 | 54 (84.4%) participants were 25–35 years of age | 6 months post-recovery | Anxiety 26.6% Depression 35.9% | HADS-A/D | 5 |
| Fernández-de-las-Peñas, Gómez-Mayordomo, García-Azorín, et al., 2021 [33] | 57 (Spain) | Case control | 38/19 | Mean 56.5 (SD 17.0) | Mean 7.3 (0.6) months after hospital discharge | Depression 24.5% Anxiety 12.3% Poor sleep quality 35.1% | HADS-A/D; PSQI | 7 |
| Grover et al., 2021 [34] | 206 (India) | Cross sectional | 95/111 | Mean 36.08 (SD 13.12) | 4–6 weeks after discharge | Anxiety 24.8% Depression 23.8% PTSD 15.5% | PHQ-4; IES-R | 5 |
| Gouraud et al., 2021 [35] | 100 (France) | Cohort | 29/71 | Median 60 (49.5–71.5) | 1 month after admission | Anxiety 31.6% Depression 22.5% | HADS-A/D; designed interview | 7 |
| O' Sullivan et al., 2021 [36] | 155 (UK) | Cohort | 28/127 | Median 39 | 3 months after acute COVID-19 illness | Anxiety/mood disorder 39.4% Sleep disturbance 12.3% | Designated tools | 5 |

Note: BDI = Beck Depression Inventory; DASS = Depression Anxiety and Stress Scale; DTS = Davidson Trauma Scale; EQ-5D-5L = EuroQuality of life-5 dimension 5 level; GAD-7 = General Anxiety Disorder 7 items; HADS = Hospital Anxiety and Depression Scale; HDRS = Hamilton Depression Rating Scale; ICD = International Classification of Diseases; IES-R = Impact of Event Scale–Revised; IQR = Interquartile Range; MMSE = Mini Mental State Examination; MoCA = Montreal Cognitive Assessment; Neuro-QoL = Neurological Disorders short form self-reported health measures; NR = nor reported; OCI = Obsessive Compulsive Inventory; PCL-5 = PTSD Checklist for DSM-5; PHQ-15 = Patient Health Questionnaire 15 items; PHQ-9 = Patient Health Questionnaire 9 items; PSQI = Pittsburgh Sleep Quality Index; PTS = Post Traumatic Stress; PTSD = Post-Traumatic Stress Disorder; SD = standard deviation; STAI = State Trait Anxiety Inventory; WHIIRS = Women's Health Initiative Insomnia Rating Scale; ZSDS = Zung Self-Rating Depression Scale.

50.0% [18]. The somatic symptom was found in 1 article with prevalence of 35.2% [16]. The cognitive deficit was found in 1 article with prevalence of 73.2% [23]. Then, obsessive compulsive was found in 1 article with prevalence of 26.0% [9].

Most of the studies used validated questionnaires. Measures of anxiety included: General Anxiety Disorder-7 (GAD-7) and State-Trait Anxiety Inventory (STAI). Four measures of depression were used: Patient Health Questionnaire-9 (PHQ-9), Zung Self-Rating Depression Scale (ZSDS), Hamilton Depression Rating Scale (HDRS), and Beck Depression Inventory (BDI). Two measures of combined depression and anxiety were used: Hamilton Anxiety and Depression Scale (HADS) and Depression Anxiety Stress Scale-21 (DASS-21). Two measures of PTSD were used: the Impact of Events Scale Revised (IES-R) and Davidson Trauma Scale (DTS). Two measures of sleep disturbance were used: Pittsburgh Sleep Quality Index (PSQI) and the Women's Health Initiative Insomnia Rating Scale (WHIIRS). The cognitive deficit was assessed by using Montreal Cognitive Assessment (MoCA). Then, Patient Health Questionnaire-15 (PHQ-15) was used to assess somatic symptoms. Some studies also used standardized questionnaires to assess the psychiatric symptoms of the participants, such as sleep difficulties, anxiety, and depression.

## Risk factors for depression

Six studies used univariate analysis to identify risk factors for depression [16,18,20,21,24,25]. Being female was associated with depression in four studies [16,20,21,24]. In contrast, studies by Menges et al. [22] and Grover et al. [34] found that there were not any correlations between sex and depression. Age was not associated with depression in six studies [16,22,23,34,36], but Taquet et al. [21] found a correlation between young age and depression. Medical history of psychiatric diagnosis and treatment was found to be associated with depression [16]. Laboratory results, including neutrophil lymphocyte ratio (NLR) upon admission [16], higher interleukin-6 (IL-6), and higher C-reactive protein (CRP) [25], were associated with depression. Then, some variables related to COVID-19, including loss of family member due to COVID-19, self-perception of the COVID-19 severity, persistent COVID-19 symptoms [16], moderate COVID-19 severity group (defined by having fever, respiratory symptoms, and imaging findings of pneumonia) [18], and hospitalization [20], were found to be associated with depression.

Four studies used multivariate analysis in assessing risk factors for depression [16,22,26,28] (Table 2). There were three major variables: sociodemographic, medical history, and COVID-19 related variables. Mazza et al. [9] reported that being female, having a previous psychiatric diagnosis, and presenting depression in the first month after being infected by SARS-CoV-2 were the risk factors for depression in the third month. Non-significant risk factors are reported in the S1 Table.

## Risk factors for anxiety

Six studies used univariate analysis to identify risk factors for anxiety [16,18–21,24]. Being female was found to be related to anxiety in Long COVID participants in 5 studies [16,19–21,24]. In contrast, one study found that sex does not correlate with anxiety [34]. Four studies rejected age as a risk factor for anxiety [16,31,34,36], but one study showed young age as one of the risk factors [21]. Other risk factors were medical history variables such as a history of psychiatric diagnosis and treatment [16]. Then, several variables related to COVID-19 were also associated with anxiety, including self-perception of the COVID-19 severity, persistent COVID-19 symptoms, history of family member infected by COVID-19, follow-up time [16], and moderate COVID-19 severity group [18].

Three studies used multivariate analysis to assess anxiety risk factors [16,26,28] (Table 2). There were three major variables: sociodemographic, medical history, and COVID-19 related variables. Some risk factors that were non-significant in several reviewed studies are reported in the S1 Table.

**Table 2. Risk factors for psychiatric symptoms among Long COVID participants from multivariate analysis.**

| Authors | Identified risk factors |
|---|---|
| **Depression** | |
| Huarcaya-Victoria et al., 2021 [16] | Female (PRa 2.11 [1.16–3.84])<br>History of psychiatry treatment (PRa 2.29 [1.10–4.47])<br>Loss of family due to COVID-19 (PRa 2.00 [1.12–3.58])<br>Perception of the COVID-19 severity (PRa 6.90 [2.09–22.78])<br>At least 1 LongCOVID-19 symptom (PRa 7.80 [2.16–28.15])<br>Disease severity scale 5–6 (OR 1.77 [1.05–2.97]) |
| C. Huang et al., 2020 [26] | Female (OR 1.80 [1.39–2.34])<br>Disease severity scale 5–6 (OR 1.77 [1.05–2.97]) |
| Fernández-de-Las-Peñas, Pellicer-Valero, et al., 2021 [28] | Female (OR 2.14 [1.25–3.65])<br>Days at hospital (OR 1.05 [1.035–1.077])<br>Onset of dyspnea at hospital admission (OR 4.86 [3.01–7.85])<br>Onset of myalgia at hospital admission (OR 1.74 [1.08–2.78])<br>The number of COVID-19 symptoms at hospital admission (OR 2.96, [1.80–4.85]) |
| Menges et al., 2021 [22] | Lower educational status (OR 0.31 [0.11–0.85])<br>Being unemployed (OR 2.53 [1.12–5.63]) |
| **Anxiety** | |
| Huarcaya-Victoria et al., 2021 [16] | Female (PRa 2.70 [1.31–5.57], p = 0.007)<br>Live with father and/or other family members (PRa 6.64 [1.06–41.52], p = 0.043)<br>History of psychiatric diagnosis (PRa 2.84 [1.25–6.46], p = 0.013)<br>History of psychiatric treatment (PRa 3.50 [1.56–7.87], p = 0.002)<br>At least 1 persistent COVID-19 symptoms (PRa 11.50 [3.07–43.15], p<0.001) |
| C. Huang et al., 2020 [26] | Female (OR 1.80 [1.39–2.34])<br>Disease severity (OR 1.77 [1.05–2.97]) |
| Fernández-de-Las-Peñas, Pellicer-Valero, et al., 2021 [28] | Female (OR 3.11 [1.745.54])<br>The number of COVID-19 symptoms at hospital admission (OR 3.21 [1.87–5.51])<br>Days at hospital (OR 1.05 [1.025–1.071])<br>Onset of dyspnea at hospital admission (OR 4.22 [2.50–7.10])<br>Onset of myalgia at hospital admission (OR 2.85 [1.70–4.79]) |
| **PTSD** | |
| Bellan et al., 2021 [27] | Male (OR 0.34 [0.14–0.84]) |
| Huarcaya-Victoria et al., 2021 [16] | Educational degree of secondary school (PRa 0.36 [0.15–0.86])<br>History of psychiatric diagnosis (PRa 3.19 [1.49–6.80])<br>History of psychiatric treatment (PRa 4.02 [1.93–8.39])<br>Self-perception of the COVID-19 severity (PRa 12.94 [1.86–90.07])<br>At least 1 persistent COVID-19 symptoms (PRa 17.84 [2.07–153.58]) |
| De Lorenzo et al., 2021 [29] | Comorbidities psychiatric disorder (OR 10.69 [2.09–78.75])<br>Insomnia in month 1 (OR 19.41 [3.44–180.26])<br>Anxiety in month 1 (OR 5.45 [1.12–31.00])<br>PTSD in month 1 (OR 4.81 [1.03–24.04]) |
| **Somatic symptoms** | |
| Huarcaya-Victoria et al., 2021 [16] | Female (PRa 1.90 [1.14–3.18])<br>History of psychiatric diagnosis (PRa 2.57 [1.47–4.50])<br>History of psychiatric treatment (PRa 2.78 [1.57–4.92])<br>Loss of family due to COVID-19 (PRa 1.69 [1.01–2.82])<br>At least 1 persistent COVID-19 symptoms (PRa 6.17 [2.48–15.35]) |
| Shang et al., 2021 [30] | Female (HR 1.569 [1.126–2.188]) |
| Fernández-de-Las-Peñas, Torres-Macho, et al., 2021 [17] | Obesity (OR 2.10 [1.13–3.83]) |
| **Cognitive deficits** | |

*(Continued)*

**Table 2.** (Continued)

| Authors | Identified risk factors |
|---|---|
| Frontera et al., 2021 [31] | Age (OR 1.03 [1.01–1.05])<br>Race (white) (OR 0.41 [0.22–0.78])<br>Education > 12 years (OR 0.40 [0.18–0.89])<br>History of dementia (OR 4.48 [1.16–12.37]) |
| Goraud et al., 2021 [35] | Age (OR 1.05 [1.01–1.09])<br>HADS score (OR 1.96 [1.08–3.57])<br>ICU admission (OR 0.22 [0.05–0.90]) |

Note: CI = confident interval; COVID-19 = corona virus disease 2019; HADS = Hospital Anxiety and Depression Scale; HR = hazard ratio; OR = odds ratio; PR = prevalence ratio.

## Risk factors for PTSD

One study used univariate analysis in assessing risk factors for PTSD in Long COVID participants [16]. Medical history of psychiatric diagnosis and treatment was found to be one of the risk factors [16]. In addition, self-perception of the COVID-19 severity, persistent COVID-19 symptoms, and follow-up time were also associated with PTSD [16].

Three studies used multivariate analysis in assessing risk factors for PTSD [16,27,29] (Table 2). Being male was found to be associated with PTSD in one study [27]. However, three studies stated that gender did not correlate with PTSD [16,29,34]. Some risk factors that were non-significant in several reviewed studies are reported in the S1 Table.

## Risk factors for somatic symptoms

One study used univariate analysis to assess risk factors for somatic symptoms. Somatic symptoms were assessed using PHQ-15 [16]. It evaluated physical symptoms experienced by the patients, including stomach pain, back pain, headaches, and other physical symptoms. Huarcaya-Victoria et al. [16] revealed that female gender, history of psychiatric diagnosis and treatment, loss of family member due to COVID-19, and having persistent COVID-19 were associated with somatic symptoms. The same study also examined the risk factors using multivariate analysis, supporting that being female, having a history of psychiatric diagnosis and treatment, loss of a family member due to COVID-19, and having at least one persistent COVID-19 symptom as predictors for somatic symptoms in long COVID participants (Table 2).

## Risk factors for sleep disturbances, poor sleep quality, and insomnia

Three studies used univariate analysis to identify risk factors for sleep disturbances in long COVID participants [17–19]. Those risk factors included being female [19], having a medical history of obesity [17], and having moderate COVID-19 severity [18]. In multivariate analysis, only two studies assessed the risk factors (Table 2). Being female [30] and obese [17] was associated with sleep disturbances, poor sleep quality, and insomnia. However, Romero-Duarte et al. [24] found that gender does not affect sleep.

## Risk factors for cognitive deficits

Two studies assessed risk factors for cognitive deficit by multivariate analysis (Table 2). The predictive factors included sociodemographic, medical history, and COVID-19 related variables. Besides those risk factors, several studies also reported non-significant risk factors, including gender [35], marital status, presence or absence of physical illness, duration of

hospital or ICU stay, days since discharge [34], and neurological complications during hospitalization [31].

## Discussion

This systematic review discovered that among Long COVID participants, psychiatric problems like anxiety, depression, sleeping problems, and PTSD symptoms were prevalent. Most of the instruments used in the studies were standardized instruments, thereby providing more valid and accurate diagnoses. Of all the psychiatric symptoms found in studies, anxiety was the most prevalent symptom in participants with Long COVID, followed by depression, sleep difficulties, and PTSD. The associated risk factors mostly found for the psychiatric symptom in this population were being female and previous psychiatric diagnosis. Meanwhile, cognitive deficits, obsessive-compulsive and somatic symptoms were the least reported.

These findings were similar to a meta-analysis showing that anxiety, depression, and sleep difficulties are the most prevalent psychiatric symptoms in acute COVID-19 participants [37]. The results of our current study were concurrent with several others. In a follow-up study of post COVID participants by Chevinsky et al. [38], anxiety was the most common psychiatric symptom seen in 30–60 days post COVID participants. Most studies also discussed that being female is the risk factor for anxiety associated with Long COVID. Seens et al. [39] stated that this might happen due to the feminine tendency in mental health toward internalizing disorders. In addition, women are more accustomed to interactions and social support outside of the household for maintaining mental health. Therefore, social isolation might have a negative impact on the female gender [39].

Besides anxiety, being female was also one of the risk factors associated with depression among Long COVID-19 participants. During the COVID-19 pandemic, females tend to have more symptoms of hyperactivity, negative cognitive and mood disturbances, which consequently could lead to the development of depression [16]. A study by Taquet et al. even showed that the depressive symptoms among female participants significantly worsened over time, and COVID-19 survivors remain clinically depressed three months after hospital discharge [21].

Sleep difficulties, poor sleep quality, and insomnia were also commonly seen. In addition to being female, obesity was also one of the risk factors for sleep difficulties. Obesity is a chronic multifactorial metabolic disease that could have contributed to multiple post COVID-19 symptoms, including sleep difficulties and poor sleep quality, mainly due to obesity-related hyperinflammation, immune dysfunction, and co-morbidities [17,40].

Development of PTSD symptoms were more common is those with previous psychiatric diagnosis and comorbidities with other psychiatric disorders. Previous research shows that Long COVID participants with higher PTSD scores also had higher anxiety and depressive scores and reported higher levels of fatigue, stigma, and cognitive deficits [34]. PTSD is often reported during the COVID pandemic due to indirect consequences of living under stress, uncertainty, and changes in daily life rather than due to the disease itself [6].

Besides recognizing the Long COVID symptoms, it is also necessary to understand the pathophysiology of Long COVID. Long COVID symptoms is believed to be linked with damage of blood-brain barrier, neurotransmission dysregulation, organ dysfunction (lung, liver, kidney), negative social and psychological factors, etc [23,41]. Several studies suggest that neuropsychiatric disorders on Long COVID patients may be linked to an inflammatory process that is overactive and has high levels of pro-inflammatory cytokines such as TNF-α, IL-6, IL-2, IL-7, and granulocyte-colony stimulating factor that remains elevated after the acute infection [40]. Furthermore, a significant number of antinuclear antibodies in people with Long

COVID supports an autoimmune origin of the neurocognitive deficits [42]. COVID-19 also affected brain structure which can eventually lead into further neuropsychiatric disorder. A study found 71% of people with Long COVID had abnormalities on magnetic resonance imaging (MRI) mainly in the white matter and 46% presented with impaired neurocognitive function at four months follow-up after discharge [43].

Another point to be highlighted is the possibility of physical symptoms effects on the mental status in this population. Mental health issues in Long COVID patients were known to be associated with persistent physical symptoms, such as myalgia and shortness of breath. This may be bidirectional. The physical symptoms could result in psychiatric symptoms and the psychiatric symptoms may show as physical symptoms [12].

Even though previous systematic reviews about factors associated with post Covid sequelae have been done, our study focused primarily on Long Covid sequelae. Therefore, we only included participants with psychiatric symptoms that persisted four weeks after the initial infection, according to the NICE definition of Long COVID [8]. This study proved that psychiatric complications of COVID-19 persisted even after a long time, which could become a major public health burden for COVID survivors. This condition should be regarded as the potential cause of a delayed pandemic in the medium to long term [8]. Therefore, it is recommended to closely monitor people experiencing Long COVID in the long term.

## Limitations

First, we could not perform a meta-analysis of the factors associated with psychiatric symptoms in Long COVID patients (e.g., a meta-analysis) due to the heterogeneity of studies' outcomes and designs. However, our main interest was identifying psychiatric symptoms in Long COVID-19 patients and risk factors associated with developing those symptoms. Consequently, we had to compromise on the quality of included studies and on the ability to rigorously estimate the risk factors of psychiatric symptoms in Long COVID patients.

Several studies in our review were cross-sectional and retrospective case-control studies, which could lead to a higher risk of recall bias. Furthermore, the length of follow up is stated in Table 1. However, we did not specifically address the influence of the difference in the length of follow up between the studies, as the impact of these differences can only be speculated without a meta-analysis. In addition, our review was limited to articles published in English, which could have resulted in selection bias.

Furthermore, we found a wide range of prevalence of psychiatric symptoms between studies. The use of different instruments to assess the symptoms was potentially a major huge factor of heterogeneity in the prevalence. In addition, the difference in follow-up times between studies was a contributor to the prevalence variation. Moreover, almost all of the included studies in this review used self-report measurements. Participants may over- or under-report their psychiatric symptoms, which could directly or indirectly result in reporting bias. Future research with a prospective design involving developing countries is recommended. Assessment of psychiatric symptoms by mental health professionals is preferable to reduce reporting and social desirability bias.

## Conclusion

In summary, this review found that psychiatric symptoms such as anxiety, depression, sleep difficulties, and PTSD symptoms were common among Long COVID participants. Being female and having previous psychiatric diagnoses were risk factors for developing those psychiatric symptoms. Detailed screening for mental disorders and early intervention in those groups will hopefully improve the quality of life of patients with Long COVID-19.

## Supporting information

**S1 Checklist. PRISMA 2020 checklist.**
(DOCX)

**S1 Table. Full non-significant risk factors.**
(DOCX)

**S1 File. Full process of study selection.**
(XLSX)

**S2 File. Newcastle-Ottawa Scale.**
(XLSX)

## Author Contributions

**Conceptualization:** Shelly Iskandar.

**Investigation:** Halwa Zakia.

**Supervision:** Shelly Iskandar.

**Writing – original draft:** Halwa Zakia, Kent Pradana, Shelly Iskandar.

**Writing – review & editing:** Halwa Zakia, Kent Pradana, Shelly Iskandar.

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
