## [Decision Letter · Decision Letter 0]

23 Aug 2022

PONE-D-22-05703Risk Factors for Psychiatric Symptoms in Patients with Long COVID: A Systematic ReviewPLOS ONE

Dear Dr. Iskandar,

Thank you for submitting your manuscript to PLOS ONE. After careful consideration, we feel that it has merit but does not fully meet PLOS ONE’s publication criteria as it currently stands. Therefore, we invite you to submit a revised version of the manuscript that addresses the points raised during the review process. The reviewer have made several comments about the language and method. Please follow the instruction from the reviewer regarding the method section.Please make the revision. 

We look forward to receiving your revised manuscript.

Kind regards,

Rizaldy Taslim Pinzon

Academic Editor

PLOS ONE

https://journals.plos.org/plosone/s/file?id=ba62/PLOSOne_formatting_sample_title_authors_affiliations.pdf".

Reviewers' comments:

Reviewer's Responses to Questions

**Comments to the Author**

1. Is the manuscript technically sound, and do the data support the conclusions?

Reviewer #1: Yes

2. Has the statistical analysis been performed appropriately and rigorously? 

Reviewer #1: N/A

3. Have the authors made all data underlying the findings in their manuscript fully available?

Reviewer #1: Yes

4. Is the manuscript presented in an intelligible fashion and written in standard English?

Reviewer #1: No

5. Review Comments to the Author

Reviewer #1: Major comments: This is an interesting systematic review about risk factors of psychiatriuc symptoms in patients with long COVID.

However, some point should be improved. The English language should be checked by a native English speaker. Also, the authors should be more precise especially when citing references, because in the introduction section many citations don’t match the purpose for which they are cited.

Specific comments

Introduction section

Page 3 :

Please correct your statement about reference 6, since it is not dealing with COVID-19 participants, but with general population during COVID-19, whoch is very different from what you are dealing in the following sentence about reference 7.

Page 4: again, the citation of reference 9 is false, this reference is A meta-analysis dealing with acute psychiatric signs of coronaviruses and not persistence after recovery.

Results section:

The term “somatic symptoms” is unclear to me, since only one study has evaluated this dimension, some more data about the nature of “somatic symptoms” may be given by the author for informative purpose.

In each section, you should not cite the unsignificant risk factors for each pathology. They should be mentioned in your table S1 and not in the text. It is better to focus on positive results and contradictory results between studies.

Discussion section

This section is too redundant compared to the result section. You should not repeat the entire results, but rather discuss more deeply some points, maybe raising pathophysiological hypothesis. You should also stress on what is new in your study since the last systematic review on the subject published in October 2021

Figure 1: please improve the quality of the figure, I can’t read it in its current state.

6. PLOS authors have the option to publish the peer review history of their article (what does this mean?). If published, this will include your full peer review and any attached files.

Reviewer #1: No

---

## [Author Response · Author response to Decision Letter 0]

30 Aug 2022

Dear editor and reviewer, 

We thank the editor and reviewer for your valuable comments and suggestions. We carefully reviewed your comments/suggestions and revised the manuscript accordingly.

Revisions are highlighted using the yellow color in the text.

https://journals.plos.org/plosone/s/file?id=ba62/PLOSOne_formatting_sample_title_authors_affiliations.pdf".

• Thank you very much for the input. We have revised the manuscript according to PLOS ONE’s style requirements and templates, according to your suggestion.

Reviewers' comments:

Comments to the Author

Review Comments to the Author

Reviewer #1: Major comments: This is an interesting systematic review about risk factors of psychiatric symptoms in patients with long COVID.

However, some point should be improved. The English language should be checked by a native English speaker. Also, the authors should be more precise especially when citing references, because in the introduction section many citations don’t match the purpose for which they are cited.

• Thank you very much for your consideration and advice. We had the manuscript checked and edited by our academic proofreader (certificate attached). We also carefully re-checked and revised the citations in our manuscript.

Specific comments

Introduction section

Page 3:

Please correct your statement about reference 6, since it is not dealing with COVID-19 participants, but with general population during COVID-19, which is very different from what you are dealing in the following sentence about reference 7.

• Thank you very much for your input, we have changed the references and statement in order to provide more clarity on the topic. Changes were made on page 3, line 44-46.

Lines 44-46 now read: “A systematic review in 2021 showed that depression, anxiety, post-traumatic stress disorder (PTSD), cognitive deficits, fatigue and sleep disturbances were commonly found in COVID-19 survivors [6]”

Page 4: again, the citation of reference 9 is false, this reference is A meta-analysis dealing with acute psychiatric signs of coronaviruses and not persistence after recovery.

• Thank you very much for your correction, we have changed the reference 9. We have added “A preliminary data suggested that some psychiatric disorder such as anxiety and depression persisted in patients who had already recovered from COVID-19 [19].” on page 3, line 58-59.

Results section:

The term “somatic symptoms” is unclear to me, since only one study has evaluated this dimension, some more data about the nature of “somatic symptoms” may be given by the author for informative purpose.

• We have added more information about somatic symptoms in line 255-257.

Lines 255-257 now read: “Somatic symptoms were assessed using PHQ-15 [17]. It evaluated physical symptoms experienced by the patients, including stomach pain, back pain, headaches, and other physical symptoms.”

In each section, you should not cite the unsignificant risk factors for each pathology. They should be mentioned in your table S1 and not in the text. It is better to focus on positive results and contradictory results between studies.

• Thank you. We have erased statements about unsignificant risk factors, and listed them in Table S1, following your suggestion.

Discussion section

This section is too redundant compared to the result section. You should not repeat the entire results, but rather discuss more deeply some points, maybe raising pathophysiological hypothesis. You should also stress on what is new in your study since the last systematic review on the subject published in October 2021

• We’ve revised the discussion section according to your suggestion. We’ve also added some patophysiological hypothesis in the discussion section.

Lines 301-304 now read: “Anxiety is the most common psychiatric symptoms seen in Long COVID individuals in our study. Psychiatric complications due to COVID-19 could happen both by immune response to the coronavirus, or due to psychological stressors during pandemic such as social isolation, concerns about infecting others, and stigma [9]”

Lines 308-311 now read: “Seens et al., stated that this happens due to the feminine tendency in mental health toward internalizing disorders. In addition, women are more accustomed to interactions and social support outside of the household for maintaining mental health, therefore, social isolation might have a negative impact on female gender [42]”

Lines 318-321 now read: “A meta-analysis showed that inflammatory cytokines such as interleukin-6, TNF-alpha, and interleukin-10 were positively associated with depression, studies further showed that those cytokines were associated with post COVID sequelae and those inflammatory cytokines stays elevated even until 8 months post COVID infection [43, 44]”

Figure 1: please improve the quality of the figure, I can’t read it in its current state.

• Thank you for the input, we have improved the quality of the figure according to your input. We also have used PACE provided by the PLOS ONE journal to suit the format of the figure.

---

## [Editor Report · Decision Letter 1]

21 Sep 2022

PONE-D-22-05703R1Risk Factors for Psychiatric Symptoms in Patients with Long COVID: A Systematic ReviewPLOS ONE

Dear Dr. Iskandar,

Thank you for submitting your manuscript to PLOS ONE. After careful consideration, we feel that it has merit but does not fully meet PLOS ONE’s publication criteria as it currently stands. Therefore, we invite you to submit a revised version of the manuscript that addresses the points raised during the review process. Thank you for your prompt reply. In the discussion section please mention, why you do not performed meta analysis for combing the data? Please focus the limitation of your study on the methodological aspects and the quality of the obtained studies. 

Please submit your revised manuscript by Nov 05 2022 11:59PM. If you will need more time than this to complete your revisions, please reply to this message or contact the journal office at plosone@plos.org. Please include the following items when submitting your revised manuscript:A rebuttal letter that responds to each point raised by the academic editor and reviewer(s). You should upload this letter as a separate file labeled 'Response to Reviewers'.A marked-up copy of your manuscript that highlights changes made to the original version. You should upload this as a separate file labeled 'Revised Manuscript with Track Changes'.An unmarked version of your revised paper without tracked changes. You should upload this as a separate file labeled 'Manuscript'.

We look forward to receiving your revised manuscript.

Kind regards,

Rizaldy Taslim Pinzon

Academic Editor

PLOS ONE

Journal Requirements:

Additional Editor Comments:

Thank you for your prompt responses. There are many studies that have been obtained. Do you consider to perform meta analysis for combining the data ? The limitations of your review should focus on the methodological flaws and the availability and quality of the evidences.

---

## [Author Response · Author response to Decision Letter 1]

9 Oct 2022

Response to Editor

[Comment 1] There are many studies that have been obtained. Do you consider to perform meta-analysis for combining the data? 

Response: Thank you for your feedback and suggestion. Yes, there are many studies that have been retrieved. However, due to the significant risk of bias present in the included papers, we did not perform meta-analyses. This might result in inaccuracies and a false outcome that appears to have more credibility.

[Comment 2] The limitations of your review should focus on the methodological flaws and the availability and quality of the evidences.

Response: Thank you for pointing this out. We have added more information about the the limitation on the methodological. Changes were made on page 22-23, line 340-359.

Lines 333-338 now read: “Some limitations were noted in this systematic review. First, we were unable to perform a meta-analysis of the factors associated with psychiatric symptoms in Long COVID patients due to the heterogeneity of studies’ outcomes and designs. However, our main interest was to identify psychiatric symptoms in Long COVID-19 patients and risk factors associated with the development of those symptoms. Consequently, we had to compromise on the quality of included studies and on the ability to rigorously estimate the risk factors of psychiatric symptoms in Long COVID patients…”

---

## [Editor Report · Decision Letter 2]

11 Oct 2022

PONE-D-22-05703R2Risk Factors for Psychiatric Symptoms in Patients with Long COVID: A Systematic ReviewPLOS ONE

Dear Dr. Iskandar,

Thank you for submitting your manuscript to PLOS ONE. After careful consideration, we feel that it has merit but does not fully meet PLOS ONE’s publication criteria as it currently stands. Therefore, we invite you to submit a revised version of the manuscript that addresses the points raised during the review process.

 - Some grammatical error still present. - Table 2 is not standard for reporting the systematic review. Please use common ways to report the outcome.- Please follow PRISMA guideline. - The prevalence of psychiatric symptoms are vary between studies. Please try to explore the causes. Is it because of different diagnosis tools? Is it because of different patient characteristics ?

We look forward to receiving your revised manuscript.

Kind regards,

Rizaldy Taslim Pinzon

Academic Editor

PLOS ONE

---

## [Author Response · Author response to Decision Letter 2]

11 Dec 2022

1. Some grammatical error still present. 

Response: Thank you for your comment. We have carefully corrected the grammatical error.

2. Table 2 is not standard for reporting the systematic review. Please use common ways to report the outcome.

Response: Thank you for pointing this out. We have updated the table to more desirable form.

3. Please follow PRISMA guideline. 

Response: Thank you for your kind reminder. We have followed PRISMA guideline on writing this review. An additional file of PRISMA guideline checklist is attached in this resubmission.

4. The prevalence of psychiatric symptoms are vary between studies. Please try to explore the causes. Is it because of different diagnosis tools? Is it because of different patient characteristics ?

Response: Thank you for your input, we have added more information of the explanation in wide range of prevalence found in our review. Changes were made on page 24-25, line 356-359.

“Furthermore, we found a wide range of prevalence of psychiatric symptoms between studies. The use of different instruments to assess the symptoms was potentially a major huge factor of heterogeneity in the prevalence. In addition, the difference in follow-up times between studies was a contributor in the prevalence variation.”

---

## [Decision Letter · Decision Letter 3]

20 Feb 2023

PONE-D-22-05703R3Risk Factors for Psychiatric Symptoms in Patients with Long COVID: A Systematic ReviewPLOS ONE

Dear Dr. Iskandar,

Thank you for submitting your manuscript to PLOS ONE. After careful consideration, we feel that it has merit but does not fully meet PLOS ONE’s publication criteria as it currently stands. Therefore, we invite you to submit a revised version of the manuscript that addresses the points raised during the review process.

please address the last minor changes requested by the reviewer

We look forward to receiving your revised manuscript.

Kind regards,

Andrea Martinuzzi

Academic Editor

PLOS ONE

Journal Requirements:

Reviewers' comments:

Reviewer's Responses to Questions

**Comments to the Author**

1. If the authors have adequately addressed your comments raised in a previous round of review and you feel that this manuscript is now acceptable for publication, you may indicate that here to bypass the “Comments to the Author” section, enter your conflict of interest statement in the “Confidential to Editor” section, and submit your "Accept" recommendation.

Reviewer #1: All comments have been addressed

Reviewer #2: All comments have been addressed

2. Is the manuscript technically sound, and do the data support the conclusions?

Reviewer #1: Yes

Reviewer #2: Yes

3. Has the statistical analysis been performed appropriately and rigorously? 

Reviewer #1: Yes

Reviewer #2: Yes

4. Have the authors made all data underlying the findings in their manuscript fully available?

Reviewer #1: Yes

Reviewer #2: Yes

5. Is the manuscript presented in an intelligible fashion and written in standard English?

Reviewer #1: Yes

Reviewer #2: Yes

6. Review Comments to the Author

Reviewer #1: The quality of the MS has significantly improved. The authors provided supplementary materials with available data. The english has also been revised.

Reviewer #2: The figure quality of PRISMA should be improved. Please revise table 2. It is too long. You should mention "risk factors" not statistical analysis for independent risk factors.

Begin the discussion with the finding of your review.

7. PLOS authors have the option to publish the peer review history of their article (what does this mean?). If published, this will include your full peer review and any attached files.

Reviewer #1: No

Reviewer #2: **Yes: **Rizaldy Taslim Pinzon

---

## [Author Response · Author response to Decision Letter 3]

6 Mar 2023

Dear editor and reviewer, 

We thank the editor and reviewer for your valuable comments and suggestions. We carefully reviewed your comments/suggestions and revised the manuscript accordingly.

Revisions are tracked using track changes feature of the Microsoft Word.

Reviewer #1: The quality of the MS has significantly improved. The authors provided supplementary materials with available data. The English has also been revised.

Reviewer #2: The figure quality of PRISMA should be improved. Please revise table 2. It is too long. You should mention "risk factors" not statistical analysis for independent risk factors. Begin the discussion with the finding of your review.

Response: Thank you for your response. We have increased the quality of the PRISMA flow chart figure. We have shortened the Table 2 (we remove the p-value). The discussion session now is started with the mind finding of this review (pg. 18 page 286-287) 

Note:

1. We do not deposit our laboratory/study protocols in protocols.io because we have registered our review in PROSPERO.

2. We have uploaded your figure files to the Preflight Analysis and Conversion Engine (PACE) digital diagnostic tool.

3. We have added three new paragraphs. The first one discusses the pathophysiology of Long COVID (pg 20, ln 320-ln 333). The second one discusses the bidirectionality of physical and mental health of Long COVID (pg 20, ln 334-pg 21, ln 338). The third one discusses the burden of the Long COVID (pg 21, ln 339-346).

---

## [Decision Letter · Decision Letter 4]

23 Mar 2023

Risk Factors for Psychiatric Symptoms in Patients with Long COVID: A Systematic Review

PONE-D-22-05703R4

Dear Dr. Iskandar,

We’re pleased to inform you that your manuscript has been judged scientifically suitable for publication and will be formally accepted for publication once it meets all outstanding technical requirements.

Kind regards,

Andrea Martinuzzi

Academic Editor

PLOS ONE

Additional Editor Comments (optional):

Reviewers' comments:

Reviewer's Responses to Questions

**Comments to the Author**

1. If the authors have adequately addressed your comments raised in a previous round of review and you feel that this manuscript is now acceptable for publication, you may indicate that here to bypass the “Comments to the Author” section, enter your conflict of interest statement in the “Confidential to Editor” section, and submit your "Accept" recommendation.

Reviewer #2: All comments have been addressed

2. Is the manuscript technically sound, and do the data support the conclusions?

Reviewer #2: Yes

3. Has the statistical analysis been performed appropriately and rigorously? 

Reviewer #2: Yes

4. Have the authors made all data underlying the findings in their manuscript fully available?

Reviewer #2: Yes

5. Is the manuscript presented in an intelligible fashion and written in standard English?

Reviewer #2: Yes

6. Review Comments to the Author

Reviewer #2: Thank you for the kind reply and revision. Some typing error in confidence interval has been found in table. Please revise.

7. PLOS authors have the option to publish the peer review history of their article (what does this mean?). If published, this will include your full peer review and any attached files.

Reviewer #2: **Yes: **Rizaldy Taslim Pinzon

---

## [Editor Report · Acceptance letter]

31 Mar 2023

PONE-D-22-05703R4 

Risk Factors for Psychiatric Symptoms in Patients with Long COVID: A Systematic Review 

Dear Dr. Iskandar:

I'm pleased to inform you that your manuscript has been deemed suitable for publication in PLOS ONE. Congratulations! Your manuscript is now with our production department. 

Kind regards, 

on behalf of

Dr. Andrea Martinuzzi 

Academic Editor

PLOS ONE